# Importance of Locations of Iron Ions to Elicit Cytotoxicity Induced by a Fenton-Type Reaction

**DOI:** 10.3390/cancers14153642

**Published:** 2022-07-27

**Authors:** Kintaro Igarashi, Yoshimi Shoji, Emiko Sekine-Suzuki, Megumi Ueno, Ken-ichiro Matsumoto, Ikuo Nakanishi, Koji Fukui

**Affiliations:** 1Molecular Cell Biology Laboratory, Department of Bioscience and Engineering, College of Systems Engineering and Science, Shibaura Institute of Technology, Fukasaku 307, Minuma-ku, Saitama 337-8570, Japan; n10003@shibaura-it.ac.jp (K.I.); koji@shibaura-it.ac.jp (K.F.); 2Quantitative RedOx Sensing Group, Department of Radiation Regulatory Science Research, National Institute of Radiological Sciences, Quantum Life and Medical Science Directorate, National Institutes for Quantum Science and Technology, 4-9-1 Anagawa, Inage-ku, Chiba-shi 263-8555, Japan; shoji.yoshimi@qst.go.jp (Y.S.); ueno.megumi@qst.go.jp (M.U.); 3Quantum RedOx Chemistry Team, Institute for Quantum Life Science, Quantum Life and Medical Science Directorate, National Institutes for Quantum Science and Technology, 4-9-1 Anagawa, Inage-ku, Chiba-shi 263-8555, Japan; nakanishi.ikuo@qst.go.jp; 4Human Resources Development Center, Quantum Life and Medical Science Directorate, National Institutes for Quantum Science and Technology, 4-9-1 Anagawa, Inage-ku, Chiba-shi 263-8555, Japan; sekine.emiko@qst.go.jp

**Keywords:** hydroxyl radical, Fenton reaction, iron ion, hydrogen peroxide, reactive oxygen species, cytotoxicity, rat thymocyte, plasmid DNA, molecular distribution, oxidative stress

## Abstract

**Simple Summary:**

The Fenton reaction generates the hydroxyl radical (^•^OH), which is the most reactive and toxic reactive oxygen species and widely recognized as a key player in oxidative stress. To clarify whether this highly reactive molecule travels to its biological target, the effects of the site of the Fenton reaction on cytotoxicity were investigated. Cytotoxicity induced by the Fenton reaction was affected by the distribution of iron ions surrounding and/or being incorporated into cells. Cytotoxicity was enhanced when the Fenton reaction occurred inside cells. Instead of enhancing cytotoxicity, extracellular iron ions exerted protective effects against the cytotoxicity of extracellular hydrogen peroxide in an ion concentration-dependent manner. Distance had a negative impact on the reactivity of extracellular ^•^OH and biologically effective targets. Furthermore, an assessment of plasmid DNA breakage showed that the Fenton reaction system did not effectively induce DNA breakage.

**Abstract:**

The impact of the site of the Fenton reaction, i.e., hydroxyl radical (^•^OH) generation, on cytotoxicity was investigated by estimating cell lethality in rat thymocytes. Cells were incubated with ferrous sulfate (FeSO_4_) and hydrogen peroxide (H_2_O_2_), or pre-incubated with FeSO_4_ and then H_2_O_2_ was added after medium was replaced to remove iron ions or after the medium was not replaced. Cell lethality in rat thymocytes was estimated by measuring cell sizes using flow cytometry. High extracellular concentrations of FeSO_4_ exerted protective effects against H_2_O_2_-induced cell death instead of enhancing cell lethality. The pre-incubation of cells with FeSO_4_ enhanced cell lethality induced by H_2_O_2_, whereas a pre-incubation with a high concentration of FeSO_4_ exerted protective effects. FeSO_4_ distributed extracellularly or on the surface of cells neutralized H_2_O_2_ outside cells. Cytotoxicity was only enhanced when the Fenton reaction, i.e., the generation of ^•^OH, occurred inside cells. An assessment of plasmid DNA breakage showed that ^•^OH induced by the Fenton reaction system did not break DNA. Therefore, the main target of intracellularly generated ^•^OH does not appear to be DNA.

## 1. Introduction

Reactive oxygen species (ROS) are generated from molecular oxygen (O_2_) through mitochondrial respiration [1,2,3,4]. O_2_ is then reduced by four electrons in the mitochondrial electron transport chain [4]. However, some of the O_2_ (0.1%) used in mitochondria is reduced by one electron, which results in the generation of superoxide (O_2_^•−^) [5]. In aqueous environments, such as the cytoplasm, O_2_^•−^ is equilibrated with the hydroperoxyl radical (HO_2_^•^), which is a highly oxidative species [6]. To eliminate these reactive species, O_2_^•−^ is reduced by superoxide dismutase (SOD) to less reactive species, such as hydrogen peroxide (H_2_O_2_) and O_2_ [7]. Although the reactivity of H_2_O_2_ is low at micromolar concentrations, the highly reactive hydroxyl radical (^•^OH) is produced via a reaction with H_2_O_2_ and transition metal ions, such as Fe^2+^ and/or Cu^+^ [8]. To prevent the generation of ^•^OH, H_2_O_2_ is degenerated by catalase and/or glutathione peroxidase to water [9].

Among ROS, ^•^OH is widely recognized as a key player in oxidative stress [10,11,12]. However, since highly reactive molecules, such as ^•^OH, cannot travel far in cells due to their reactivity, it currently remains unclear whether they reach their biological target before being canceled by a reaction with other molecules. Radiation-induced ^•^OH in water was previously shown to be generated at two different local concentrations, i.e., mmol/L and mol/L levels, with intermolecular distances of 4–7 and <0.8 nm, respectively [13,14,15]. These two ^•^OH generated with an intermolecular distance < 0.8 nm react and produce H_2_O_2_ in an oxygen-independent manner, whereas those generated with an intermolecular distance of 4–7 nm cannot react with each other due to their distance. A previous study demonstrated that the ratio of mmol/L ^•^OH generation was inhibited while that of mol/L ^•^OH generation was enhanced by increases in the linear energy transfer (LET) of particle radiation [13]. Therefore, a high LET carbon-ion beam more strongly promotes the oxygen-independent generation of H_2_O_2_ over the oxygen-dependent generation of H_2_O_2_ [16]. The molecular distance of ^•^OH influences its sequential reactions. Therefore, the distance between ^•^OH and its target molecule is an important factor affecting ^•^OH-induced oxidative stress.

The Fenton reaction is a well-known chemical reaction that is utilized as a source of ^•^OH [17]. It involves the reduction in H_2_O_2_ by the reduced form of transition metal ions, such as Fe^2+^ and/or Cu^+^, which generates ^•^OH and the oxidized form of metal ions (Equation (1)). Fenton originally discovered the strong oxidation of tartaric acid following the addition of H_2_O_2_ to tartaric acid solution, which contained a trace amount of Fe^2+^ salt [18,19].
H_2_O_2_ + Fe^2+^/Cu^+^ → ^•^OH + OH^−^ + Fe^3+^/Cu^2+^(1)

Haber and Weiss [20] identified the strong oxidant produced by the reaction between H_2_O_2_ and Fe^2+^ as ^•^OH. Therefore, this reaction is generally known as the Haber–Weiss reaction and generates ^•^OH through the reduction in H_2_O_2_ by O_2_^•−^ (Equation (2)).
H_2_O_2_ + O_2_^•−^ → ^•^OH + OH^−^ + O_2_(2)

After being reduced, weakly reactive H_2_O_2_ becomes highly toxic ^•^OH. The site at which the reduction in H_2_O_2_ occurs has an important impact on the effects of oxidative stress on living cells.

Thymocytes are immune cells that constitute the thymus. Rodent thymocytes exhibit a characteristic apoptotic response [21,22]. The radiation-induced shrinkage of rodent thymocytes is easily detected and measured using a flow cytometer [23]. The size of irradiated rat thymocytes was previously shown to be significantly reduced due to apoptosis and they were classified into two discrete subpopulations: normal and smaller thymocytes. This classification has been utilized to assess the radio-protective effects of planar catechin analogues [24].

Plasmid DNA breakage has been employed to estimate the extent of biological oxidative damage in vitro [25]. A plasmid is a small circular extrachromosomal double-stranded DNA structure. Plasmid DNA has three different topoisomers: linear, open circular, and supercoiled. The original supercoiled form assumes the open circular form with single-strand breaks and the linear form with more severe double-strand breaks [26].

In the present study, the effects on cytotoxicity of the molecular location at which ^•^OH is generated by the Fenton reaction were investigated. The influence of the Fenton reaction system on cytotoxicity was examined by comparing the cell death rate of rat thymocytes. The effectiveness of the Fenton reaction was evaluated based on the extent of plasmid DNA breakage. The potential cytotoxicity of ^•^OH generated by the Fenton reaction was discussed based on the sites at which Fe^2+^ and H_2_O_2_ come into contact and react with each other.

## 2. Materials and Methods

### 2.1. Chemicals

All chemicals used in the present study were of analytical grade. Deionized water (deionization by the Milli-Q system) was used to prepare phosphate-buffered saline (PBS), ferrous sulfate (FeSO_4_) stock solutions, and H_2_O_2_ stock solutions.

### 2.2. Plasmid DNA

A pBR322 plasmid DNA solution (4361 bp, 0.5 µg/µL) in TE buffer (10 mM Tris-HCl, pH 8.0, 1 mM EDTA) was purchased from Takara Bio Inc., Shiga, Japan. pBR322 was precipitated by the addition of 0.1 vol of 3 M sodium acetate and 2.5 vol of ethanol and incubated at –20 °C for 2 h. The plasmid was collected by centrifugation at 12,000 rpm at 4 °C for 15 min in MX-100 (Tomy Seiko, Co., Ltd., Tokyo, Japan), washed with 250 µL of 70% ethanol–water, and collected again by centrifugation at 12,000 rpm at 4 °C for 15 min in MX-100 (Tomy Seiko, Co., Ltd., Tokyo, Japan) followed by removal of the residual solvent. pBR322 was dried at room temperature for 5 min. A stock solution was prepared by dissolving the dry plasmid in 50 µL H_2_O and leaving it to stand at 4 °C overnight. DNA concentrations were quantified by measuring absorbance at 260 nm (molar extinction coefficient: 50 µg/mL/cm) using the NanoPhotometer N60 (Implen, Munich, Germany).

### 2.3. Animals

Healthy 8-week-old male Wistar-MS rats were supplied by Japan SLC, Inc. (Shizuoka, Japan). Animals were housed one or two per cage in climate-controlled (23 ± 1 °C and 55 ± 5% humidity), circadian rhythm-adjusted (12-h light–dark cycle) rooms and were allowed food and water ad libitum until experiments were conducted. Rats were used for experimentation at 10–15 weeks old. Experiments were approved by the Animal Use Committee of the National Institute of Radiological Sciences, Chiba, Japan.

### 2.4. Preparation of Thymocytes

Roswell Park Memorial Institute medium (RPMI1640) was purchased from Sigma-Aldrich (St. Louis, MO, USA). The thymus was surgically removed from rats. Thymocytes were squeezed out of the thymus using tweezers, placed in PBS, and passed through a mesh to disassemble single cells. A cell suspension at a density of 10 × 10^5^ cells/mL was prepared by RPMI1640 medium containing 10% fetal bovine serum (FBS).

### 2.5. Assessment of Rat Thymocyte Lethality Induced by ^•^OH

*Experiment 1:* FeSO_4_ or H_2_O_2_ was added to the thymocyte suspension in medium at a final concentration of 100, 500, or 1000 µM. Samples were incubated at room temperature for 5 min, centrifuged at 600× *g* for 5 min, re-suspended in RPMI1640 medium containing 10% FBS, and incubated at 37 °C for another 4 h.

*Experiment 2:* Thymocyte suspensions in medium containing FeSO_4_ (0, 100, or 500 µM as the final concentration) and H_2_O_2_ (100, 400, 700, or 1000 µM as the final concentration) were incubated at room temperature for 5 min, centrifuged at 600× *g* for 5 min, re-suspended in RPMI1640 medium containing 10% FBS, and incubated at 37 °C for another 4 h.

*Experiment 3:* Thymocyte suspensions in medium containing FeSO_4_ (0, 50, 500, or 1000 µM as the final concentration) were incubated at room temperature for 30 min and H_2_O_2_ was then added to the thymocyte suspension in medium to a final concentration of 100 or 500 µM. Samples were incubated at room temperature for 5 min, centrifuged at 600× *g* for 5 min, re-suspended in RPMI1640 medium containing 10% FBS, and then incubated at 37 °C for another 4 h.

*Experiment 4:* Thymocyte suspensions in medium containing FeSO_4_ (50, 500, or 1000 µM as the final concentration) were incubated at room temperature for 30 min, centrifuged at 600× *g* for 5 min, and re-suspended in fresh RPMI1640 medium containing 10% FBS with the addition of H_2_O_2_ to a final concentration of 100 or 500 µM. Samples were incubated at room temperature for 5 min, centrifuged at 600× *g* for 5 min, re-suspended in RPMI1640 medium containing 10% FBS, and incubated at 37 °C for another 4 h.

After the 4-h incubation, cell sizes were measured using the flow cytometer FACSCalibur (Becton, Dickinson and Company, Franklin Lakes, NJ, USA). The ratio of the number of shrunken cells, i.e., apoptotic cells, to the total cell number was estimated as the cell death rate (%).

### 2.6. Assessment of Plasmid DNA Breakage Induced by ^•^OH

DNA strand breakage was examined by the conversion of supercoiled pBR322 plasmid DNA to the open circular and linear forms. Reactions were performed in 10 µL (total volume) of 10 mM phosphate buffer (pH 7.4) containing 0.1 µg pBR322 DNA under experimental conditions 1–4 described below.

*Experiment 1:* H_2_O_2_ (0.1, 1, 10, or 100 mM), FeSO_4_ (0.1, 1, 10, or 100 mM), and FeCl_3_ (0.1, 1, 10, or 100 mM) were added to aqueous solution containing plasmid DNA under aerobic or hypoxic conditions at several concentrations.

*Experiment 2:* H_2_O_2_ and FeSO_4_ were added to aqueous solution containing plasmid DNA under aerobic conditions. The final concentration of FeSO_4_ was 100 or 1000 µM, and the concentration of H_2_O_2_ added was 50%, equal, or two-fold that of FeSO_4_.

*Experiment 3:* H_2_O_2_ was added to aqueous solution containing plasmid DNA under aerobic conditions. Samples were irradiated with 0.1 or 0.5 J/cm^2^ of 254 nm UV light using an UV irradiator (CL-1000 Ultraviolet Crosslinker, UVP, LLC, Upland, CA, USA).

*Experiment 4:* The addition of 1 mM H_2_O_2_ and several concentrations of caffeine, DMSO, terephthalate, or sucrose to aqueous solution containing plasmid DNA was performed under aerobic conditions. Samples were irradiated with 0.1 J/cm^2^ of 254 nm UV light.

After being incubated at room temperature for 10 min, reaction mixtures were then treated with 2 µL of a loading buffer containing 30% glycerol, 0.25% bromophenol blue, 0.25% xylene cyanol, and 1 mM EDTA loaded onto a 1% agarose gel. Gels were run at a constant voltage of 50 V for 2 h in TBE buffer, stained in 0.5 µg/mL ethidium bromide for 1 h, washed with distilled water for 30 min, visualized under a UV transilluminator, and photographed using a digital camera. Gel images were analyzed using ImageJ software (NIH, Bethesda, MD, USA) and the amount of the open circular form was quantified.

### 2.7. Estimation of the ^•^OH yield

Regarding the Fenton reaction, an aliquot of the water solution of DMPO and H_2_O_2_ was added to a microtube, followed by an aliquot of the water solution of FeSO_4_ to start the reaction. The final concentration of DMPO was 100 mM for all experiments, and the final concentrations of H_2_O_2_ and FeSO_4_ were adjusted to 500, 250, or 125 μM depending on the experiment. The time course of DMPO-OH generated in the reaction mixture was measured by an EPR spectrometer at 20-sec intervals for 4 min. Reciprocal values of the concentration of DMPO-OH were plotted versus time after starting the reaction. The initial concentration (C_0_) of DMPO-OH was estimated by extrapolating the initial linear slope to the *Y*-axis (t = 0).

Regarding the decomposition of H_2_O_2_ by UV irradiation, a reaction mixture containing 100 mM DMPO and 1.0 mM H_2_O_2_ was irradiated with 0.1 J/cm^2^ of 254 nm UV light. The time course of DMPO-OH generated in the reaction mixture was measured by the EPR spectrometer.

### 2.8. Statistical Analysis

Significant differences between values for comparison were estimated using the TTEST function in Microsoft Excel 2010. Suitable ‘tail’ and ‘type’ for the TTEST function were selected as follows. The ‘tail’ was 2 (two-tailed distribution) for stability tests because the difference between the two data groups was simply compared. The ‘type’ was 2 (equal variance) or 3 (unequal variance), which was estimated using the FTEST function, and the Student’s or Welch’s *t*-test was performed according to the ‘type’. Grades of significance were estimated by *p* < 0.05, *p* < 0.01, and *p* < 0.001.

## 3. Results and Discussion

The effects of ^•^OH induced by the Fenton reaction on living cells were evaluated by a cell survival test using rat thymocytes [23]. Figure 1 shows the results of cytotoxicity induced by exposing cells to FeSO_4_ or H_2_O_2_ alone. The administration of FeSO_4_ alone was not cytotoxic at a concentration < 1000 μM, whereas H_2_O_2_ induced cytotoxicity in a dose-dependent manner.

Lipophilic H_2_O_2_ easily entered the intracellular volume through the cell membrane during the 5-min incubation period with an extra 10–15 min for washing, i.e., centrifugation, removing the supernatant, and adding fresh cell culture medium. H_2_O_2_ may be decomposed to generate ^•^OH by a reaction with intracellular metal ions. However, iron ions may not directly enter the intracellular volume through the cell membrane during the 15–20 min used for the incubation and washing. The incubation period of 15–20 min was too short for the uptake of iron by cells. Iron ions may become cytotoxic when they accumulate at excessive amounts [27]; however, this requires a long period of time. The accumulation of iron ions in cells to a level that promotes lipid peroxidation and induces cell death is called ferroptosis [28,29].

Figure 2 shows the effects of Fenton reaction-induced ^•^OH generation on cytotoxicity when FeSO_4_ and H_2_O_2_ were simultaneously added to cell culture samples. The experiment with 0 µM FeSO_4_ is a repetition of Figure 1B; however, in this separate experiment, the addition of 400 µM H_2_O_2_ increased cell lethality (74%), which was similar to that observed with 700 µM H_2_O_2_. Cell lethality in rat thymocytes following the addition of H_2_O_2_ alone was enhanced at H_2_O_2_ concentrations of 400–500 µM and appeared to vary in a manner that was dependent on the total incubation time (15–20 min), which included the processing time for cell washing.

The results of the experiments with 0 and 100 µM FeSO_4_ were compared, and similar cell lethality was observed at the same H_2_O_2_ doses; however, cell lethality was significantly lower (*p* < 0.01) when 400 µM H_2_O_2_ was added to the 100 µM FeSO_4_ sample than with its addition to the 0 µM FeSO_4_ sample. The addition of 100 and 400 µM H_2_O_2_ with 500 µM FeSO_4_ to cell culture samples resulted in significantly lower (*p* < 0.001) cell lethality than that observed in the control. In experiments using the same H_2_O_2_ concentration, the sample treated with 500 µM FeSO_4_ showed significantly lower (*p* < 0.001) cell lethality than those treated with the other FeSO_4_ concentrations. Higher concentrations of FeSO_4_, which may have been distributed in the medium, i.e., outside cells, exerted protective rather than cytotoxic effects. This result is consistent with previous findings [30].

Extracellular Fe^2+^ ions may react with H_2_O_2_ and decompose it to ^•^OH before H_2_O_2_ reaches cells. ^•^OH generated outside cells does not reach cells due to its high reactivity. ^•^OH may immediately react with any molecule at the site of its generation and be neutralized before it reaches cells.

Figure 3 shows the effects of a pre-incubation with FeSO_4_ on Fenton reaction-induced cell death. Solid columns show the results of Experiment 3, i.e., H_2_O_2_ was added after a pre-incubation with or without FeSO_4_. When thymocytes were pre-incubated without FeSO_4_, the addition of H_2_O_2_ induced a higher percentage (approximately 40% for 100 µM and 70% for 500 µM) of cell death than that in the experiment shown in Figure 1B. Pre-incubations with several different concentrations (50, 500, and 1000 µM) of FeSO_4_ did not change the percentage of cell death induced by the addition of H_2_O_2_ from that in samples pre-incubated without (0 µM) FeSO_4_. The pre-incubation with 500 µM FeSO_4_ decreased the percentage of cell death to lower than that in samples pre-incubated with 0 or 50 µM FeSO_4_. The percentage of cell death was similar in samples pre-incubated with 1000 µM FeSO_4_ and those pre-incubated with 50 µM FeSO_4_. When 500 µM H_2_O_2_ experiments (black solid columns) were compared, the pre-incubation with 1000 µM FeSO_4_ significantly decreased cell lethality to lower than that in samples pre-incubated without (0 µM) FeSO_4_. A simultaneous exposure to 500 µM FeSO_4_ and 100 µM H_2_O_2_ resulted in significantly lower cell lethality than in the control (Figure 2), while the pre-incubation with 500 µM FeSO_4_ and later addition of 100 µM H_2_O_2_ significantly increased cell lethality to higher than that in the control (Figure 3). A previous cell incubation with FeSO_4_ appeared to increase the potential for cell lethality induced by the later addition of H_2_O_2_. In contrast, a higher extracellular concentration of FeSO_4_ exerted protective effects against H_2_O_2_-induced cell death.

The striped columns in Figure 3 show the results of Experiment 4, i.e., H_2_O_2_ was added after the removal of FeSO_4_ from pre-incubated samples. When extracellular Fe^2+^ ions were removed before the addition of H_2_O_2_ (striped columns), cell lethality was higher than in samples treated with the same dose of H_2_O_2_ with remaining extracellular Fe^2+^ ions (solid columns). When the results of 100 µM H_2_O_2_ experiments (gray striped columns) were compared, cell lethality was found to decrease in a manner that was dependent on the FeSO_4_ concentrations used in the pre-incubation. When the results of 500 µM H_2_O_2_ experiments (black striped columns) were compared, the pre-incubation with 500 µM FeSO_4_ was found to significantly increase cell lethality over that by 50 µM FeSO_4_, however; the pre-incubation with 1000 µM FeSO_4_ significantly decreased cell lethality to lower than that by 500 µM FeSO_4_. These results suggest that the higher concentration of iron ions remaining on the surface of cells neutralized H_2_O_2_ or that iron ions taken into cells at higher concentrations leaked outside cells and reacted with H_2_O_2_ before it reached cells.

Cytotoxicity induced by the Fenton reaction was affected by the distribution of iron ions surrounding cells. The available distance for ^•^OH to travel in water may be 2 nm or less [31]. In other words, for extracellular ^•^OH to induce cell death by a sufficient amount needs to be produced at a site that is within 2 nm of a cell. Therefore, extracellular iron ions, which need to be uniformly distributed in the buffer, may protect cells by neutralizing H_2_O_2_ outside cells. However, intracellular iron ions may promote H_2_O_2_-triggered cell death. To increase cytotoxicity, the generation of ^•^OH by the Fenton reaction must occur inside cells.

To effectively induce cytotoxicity, ^•^OH needs to be produced at the intracellular space. Ionizing radiation, such as photon and/or particle-ion beams, ionizes water molecules on their path independently of whether they are located inside or outside cells. Therefore, ^•^OH may be generated inside cells by ionizing radiation and may effectively induce cell death. Previous studies reported that 2 Gy of X-ray irradiation achieved approximately 40% cell lethality [23,24]. Since 0.53 µmol/L/Gy of ^•^OH was shown to be generated by X-ray irradiation [15], 2 Gy may produce 1.1 µmol/L ^•^OH in water. Ionizing radiation generates not only ^•^OH but also most types of ROS, such as H_2_O_2_, O_2_^•−^, and HO_2_^•^, inside cells directly through water radiolysis and/or sequential reactions [32]. Furthermore, the initial generation of ^•^OH by ionizing radiation is localized, with local concentrations in the mmol/L and/or mol/L ranges [15].

Figure 4 shows the results on plasmid DNA breakage induced by exposure to H_2_O_2_, Fe^2+^ ions (FeSO_4_), or Fe^3+^ ions (FeCl_3_). H_2_O_2_ alone did not induce plasmid DNA breakage at a concentration lower than 10 mM under aerobic conditions (dark column in Figure 4A). In contrast, 10 mM or higher H_2_O_2_ induced DNA breakage under aerobic conditions, even though no significance was observed. However, DNA breakage induced by H_2_O_2_ alone was strongly enhanced under hypoxic conditions, and even 1.0 mM H_2_O_2_ induced slight DNA breakage under hypoxic conditions (light column in Figure 4A). H_2_O_2_ is a more stable and less reactive species than other ROS; nevertheless, it may become more reactive under hypoxic conditions. In contrast, 0.1 or 1.0 mM Fe^2+^ induced slight DNA breakage under aerobic conditions; however, this was inhibited under hypoxic conditions (Figure 4B). The addition of 10 mM or higher Fe^2+^ strongly induced DNA breakage under both aerobic and hypoxic conditions as if supercoil form had never existed (Figure 4B). Fe^3+^ alone did not break DNA under aerobic or hypoxic conditions at concentrations lower than 1.0 mM (Figure 4C). The addition of Fe^3+^ at a concentration of 10 mM or higher induced the disassembly of DNA under aerobic or hypoxic conditions. H_2_O_2_, Fe^2+^ ions (FeSO_4_), and Fe^3+^ ions (FeCl_3_) did not induce plasmid DNA breakage by themselves at concentrations lower than 0.1 mM, except for 0.1 mM Fe^2+^, which induced slight plasmid DNA breakage.

Figure 5 shows the results on plasmid DNA breakage caused by Fenton reaction-induced ^•^OH under aerobic conditions. Regarding the simultaneous exposure of plasmid DNA to H_2_O_2_ and Fe^2+^, the percentage of DNA breakage only slightly increased when the concentrations of both H_2_O_2_ and Fe^2+^ were 1.0 mM or higher (Figure 5B). H_2_O_2_ and Fe^2+^ ions travel relatively long distances in water, and most reach the target DNA; however, the majority of ^•^OH is neutralized at its site of generation and does not reach the target. The intermolecular distances of a compound at concentrations of 0.1 and 1.0 mM in a solution were calculated as 32 and 15 nm, respectively. When 1.0 mM FeSO_4_ and 1.0 mM H_2_O_2_ are reacted in an aqueous plasmid DNA solution, ^•^OH is ideally generated with an intermolecular distance of 15 nm. The distance between plasmid DNA and the nearest ^•^OH generated may be 0–15 nm. Similarly, the reaction of 0.1 mM FeSO_4_ and an equal concentration of H_2_O_2_ may generate ^•^OH with an intermolecular distance of 32 nm. The distance between ^•^OH generated and the plasmid DNA molecule may be longer when the concentrations of FeSO_4_ and H_2_O_2_ are lower. Nevertheless, only a very small amount of ^•^OH generated reaches plasmid DNA and breaks it.

As already shown in Figure 4, 1.0 mM Fe^2+^ induced DNA breakage under aerobic conditions, and this was inhibited by the addition of H_2_O_2_ (Figure 5B). H_2_O_2_ may react with Fe^2+^ before it reaches DNA, and ^•^OH generated by the reaction of H_2_O_2_ and Fe^2+^ also does not reach DNA due to its extremely high reactivity.

H_2_O_2_ is a source of ^•^OH, and Fe^2+^ functions as a processor for the induction of highly reactive ^•^OH. Although ^•^OH is widely recognized as a key player in oxidative stress, its extremely high reactivity limits its traveling distance and reaction efficiency with the target molecule. However, Fe^2+^/Fe^3+^ ions and/or H_2_O_2_ alone are moderately reactive and may travel long distances to the target molecule.

^•^OH generated by the decomposition of H_2_O_2_ under UV irradiation may induce plasmid DNA breakage in a manner that depends on the concentration of H_2_O_2_ and UV energy (Figure 6). The irradiation of 0.1 J/cm^2^ UV to a 10 mM H_2_O_2_ sample induced the disassembly of DNA and defied rational analyses. A total of 0.5 J/cm^2^ UV to 1 mM or higher H_2_O_2_ samples also induces the disassembly of DNA. The UV irradiation of a H_2_O_2_ solution may induce the generation of two ^•^OH (Equation (3)). However, in a dilute H_2_O_2_ water solution, the majority of the ^•^OH pairs generated may be recombined and return to H_2_O_2_ (Equation (4)) due to the lack of any other molecule around the generation site.
UV ~~~> H_2_O_2_ → 2 ^•^OH(3)
^•^OH + ^•^OH → H_2_O_2_(4)

Regenerated H_2_O_2_ again travels in water close to the target molecule. ^•^OH may be generated close to and react with DNA. As a result, the UV irradiation of H_2_O_2_ solution may continuously generate two ^•^OH in close proximity to DNA molecules. The single ^•^OH generated by the Fenton reaction (Equation (1)) may be neutralized by a free electron at its site of production.

Figure 7 shows the effects of the ^•^OH scavengers, caffeine, DMSO, terephthalate, and sucrose on the inhibition of DNA breakage by UV irradiation in samples containing H_2_O_2_. The addition of a ^•^OH scavenger to the reaction mixture suppressed DNA breakage in a concentration-dependent manner. The suppression of DNA breakage by caffeine, DMSO, and sucrose was consistent with their previously reported ^•^OH-canceling abilities [33]. The ^•^OH-scavenging effects of terephthalate, which has been used as a fluorescent ^•^OH detector [34], were similar to those of DMSO. This result suggests that plasmid DNA breakage by UV irradiation in samples containing H_2_O_2_ was caused by ^•^OH.

^•^OH generated by the UV decomposition of H_2_O_2_ is localized and not uniformly distributed in the three-dimensional space of a solvent [35]. The local concentration of ^•^OH generated by the UV decomposition of H_2_O_2_ was dependent on the concentration of H_2_O_2_. Therefore, the UV irradiation of 1 mM H_2_O_2_ solution resulted in ~1 mM ^•^OH as a local concentration. However, the average concentration of ^•^OH generated by the UV decomposition of H_2_O_2_ in a large volume was markedly lower. In a previous study [35], 0.72 J/cm^2^ (=12,000 μW/cm^2^ for 1 min) of mixed wavelength UVB irradiation to 0.98 mM H_2_O_2_ gave 3.2 μM ^•^OH as the average concentration in a large volume. In the present study, the average concentration of ^•^OH generated by the UV irradiation of 1.0 mM H_2_O_2_ at 0.1 J/cm^2^ 254 nm was estimated to be 3.6 ± 0.1 μM (*n* = 3). The EPR intensity of DMPO-OH induced by UV irradiation to the sample containing 1.0 mM H_2_O_2_ and 100 mM DMPO was stable and constant for 10 min or longer and was easily and accurately quantified. Several mM levels of ^•^OH scavengers were required to suppress plasmid DNA breakage by UV irradiation in samples containing 1.0 mM H_2_O_2_ in Figure 7, which was attributed to the local concentration of ^•^OH generated in this experiment being at the mM level.

Inhomogeneous ^•^OH generation in the Fenton reaction system was also expected in the same study [35]; however, the rapid decay of DMPO-OH under coexisting Fe^2+^/Fe^3+^ ions led to difficulties with the accurate estimation of the ^•^OH yield and may have concealed the true results. In the present study, the ^•^OH yield in the Fenton reaction system was estimated using lower concentrations of reagents (<500 μM).

As shown in Equation (1), Fe^2+^ ions and H_2_O_2_ react at a ratio of 1:1. Therefore, the amount of ^•^OH generated by the Fenton reaction is dependent on a lower concentration of Fe^2+^ or H_2_O_2_. Figure 8 shows the results of the quantification of ^•^OH induced by the Fenton reaction. Since the fast decay of DMPO-OH under coexisting iron ions resulted in difficulties with the accurate estimation of initial DMPO-OH concentrations, DMPO-OH, i.e., spin-trapped ^•^OH, concentrations in Figure 8 may have been lower than predicted values, i.e., a lower concentration of Fe^2+^ or H_2_O_2_ in the reaction mixture. When the concentration of H_2_O_2_ was higher than FeSO_4_, the estimated DMPO-OH yield was 25 or 18% of the predicted value. When the concentration of H_2_O_2_ was the same or lower than FeSO_4_, the estimated DMPO-OH yield was 82, 76, or 93% of the predicted value. Therefore, approximately 0.5 or 1 mM of ^•^OH was expected under the experimental conditions in Figure 5B. However, DNA breakage was negligible. In this experimental Fenton reaction system, the maximum available amount of ^•^OH appeared to have been generated and rapidly disappeared. Therefore, most of the ^•^OH generated in the experimental Fenton reaction system did not react with other molecules before it was canceled/neutralized.

Experimental Fenton reaction-induced ^•^OH did not easily break plasmid DNA, even when 2.0 mM Fe^2+^ and 1.0 mM H_2_O_2_ reacted (Figure 5B). Only 100 µM H_2_O_2_ significantly increased cell lethality (Figure 1B). Therefore, the target of ^•^OH through the Fenton reaction, H_2_O_2_, and/or the Fe^2+^/Fe^3+^ redox pair for cell lethality does not appear to be DNA in the cell nucleus. Most notably, the results shown in Figure 4A,B demonstrated that 1 mM or a higher concentration of H_2_O_2_ or Fe^2+^ was required to break DNA under the hypoxic conditions predicted in the intracellular space. The localized generation of ^•^OH by the UV decomposition of H_2_O_2_ reported in a previous study [34] and the results shown in Figure 5B, Figure 6 and Figure 7 in the present study showed that the continuous production of 1 mM or higher of ^•^OH was required to break DNA.

H_2_O_2_ is a membrane permeable molecule that travels through cell membranes. Membrane lipids may be other targets of oxidants instead of DNA for the induction of cell death. Ferroptosis is iron-induced lipid peroxidation-dependent cell death that differs from apoptosis and necrosis [36]. Imai et al. [37] reported another type of lipid peroxidation-dependent cell death that was independent of iron, called lipoxytosis. The accumulation of lipid peroxidation in cell membranes may trigger this type of cell death, and catalytic iron, not only Fe^2+^/Fe^3+^ ions but also some chelated iron forms, in the intracellular space may enhance lipid peroxidation and ferroptosis [38]. Iron in asbestos, which is a silicate mineral fiber that sticks into cells like needles, may cause carcinogenesis [39]. This type of carcinogenesis may be induced by cells exposed to repeated iron-catalyzed Fenton-type reactions [40]. An evaluation of the effects of lipid peroxidation on cell death induced by the Fenton-type reaction is important in the future.

## 4. Conclusions

Cytotoxicity induced by the Fenton reaction was modified by the distribution of iron ions surrounding and/or entrapped in cells. The Fenton reaction, i.e., the generation of ^•^OH, needs to occur inside a cell in order to effectively induce cytotoxicity. Chemically induced ^•^OH in the Fenton reaction system does not easily cause DNA breakage because the distance that ^•^OH travels before being neutralized is very limited. Therefore, the main target of ^•^OH, which is intracellularly induced from H_2_O_2_ or other reactive species, including the Fe^2+^/Fe^3+^ redox pair, for the induction of cytotoxicity does not appear to be DNA. Iron ions must be entrapped in cells for the induction of cytotoxicity. Although Fe^2+^ ions directly induce DNA breakage, they do not have to travel to reach DNA in order to result in cell lethality.

## Figures and Tables

**Figure 1 cancers-14-03642-f001:**
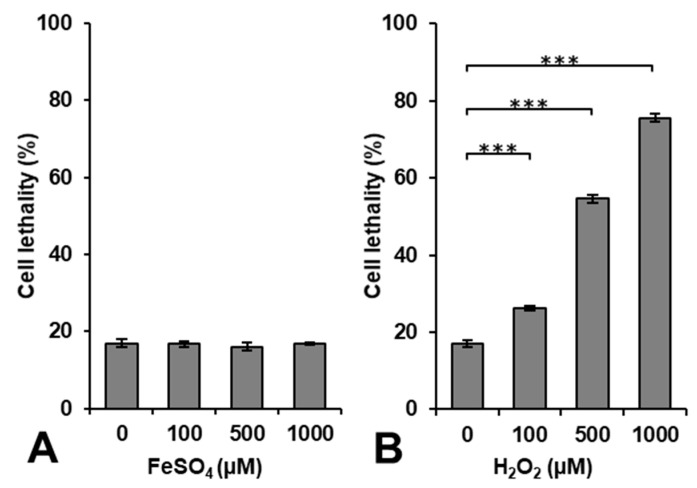
Estimation of cytotoxicity induced by the administration of H_2_O_2_ or FeSO_4_ alone. Cell lethality in rat thymocytes after a 5-min incubation with (**A**) FeSO_4_ or (**B**) H_2_O_2_. Columns and error bars indicate the average and SD of three experiments. *** indicates a significant difference of *p* < 0.001.

**Figure 2 cancers-14-03642-f002:**
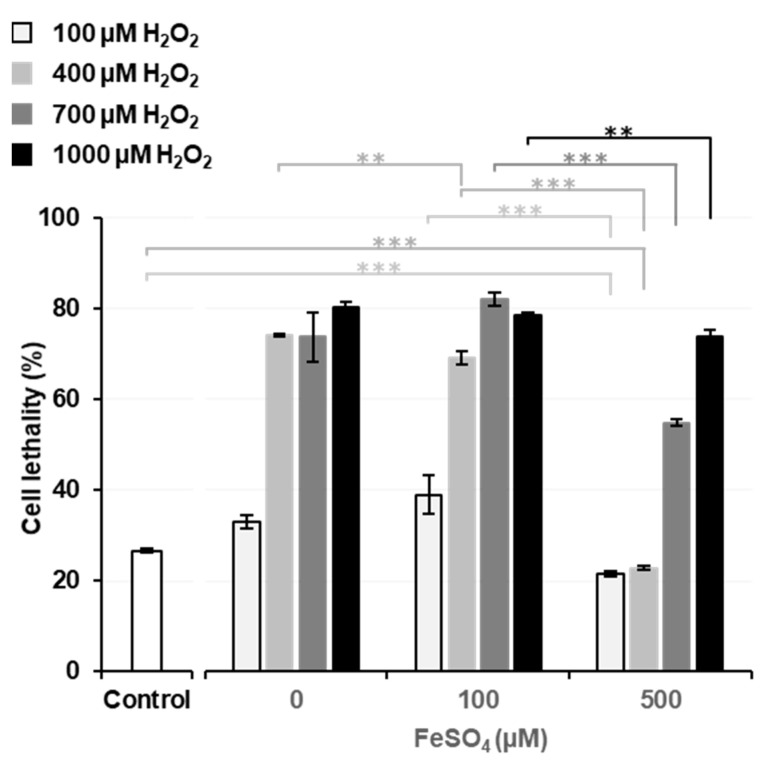
Cytotoxicity induced by the simultaneous administration of H_2_O_2_ and FeSO_4_. Neither H_2_O_2_ nor FeSO_4_ was added to the control sample, i.e., the cell suspension was just incubated. Columns and error bars indicate the average and SD of three experiments. All columns are significantly different (*p* < 0.05 or less) from the control. When experiments without FeSO_4_ were compared, no significance differences were observed between 400 and 700 µM H_2_O_2_ or between 700 and 1000 µM H_2_O_2_. When experiments with 500 µM FeSO_4_ were compared, no significant differences were noted between 100 and 400 µM H_2_O_2_. Other significant differences are indicated in the figure as ** and ***, which correspond to *p* < 0.01 and *p* < 0.001, respectively.

**Figure 3 cancers-14-03642-f003:**
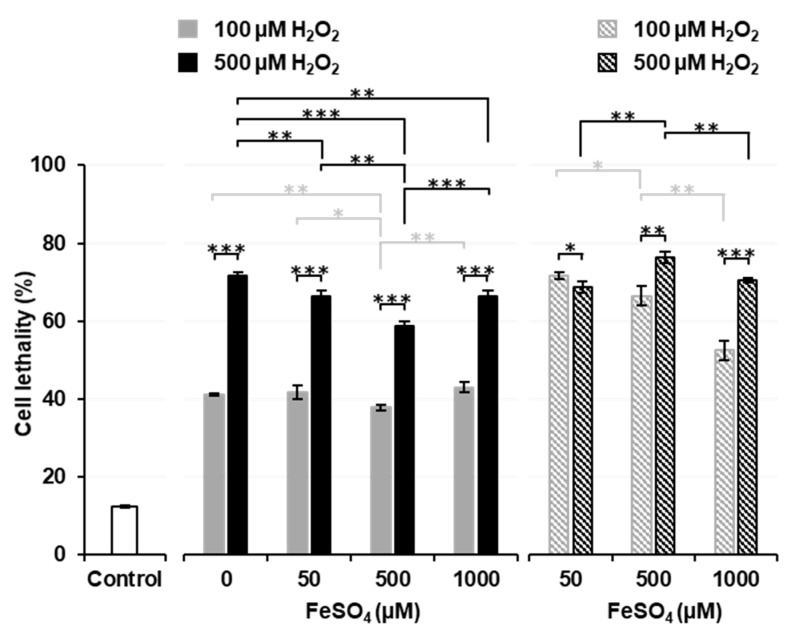
Effects of the pre-incubation of cells with FeSO_4_ on cytotoxicity induced by the administration of H_2_O_2_. Neither H_2_O_2_ nor FeSO_4_ was added to the control sample, i.e., the cell suspension was just incubated. Solid columns show the results of Experiment 3, i.e., H_2_O_2_ was added after the pre-incubation with or without FeSO_4_. Striped columns show the results of Experiment 4, i.e., H_2_O_2_ was added after the removal of FeSO_4_ from pre-incubated samples. Columns and error bars indicate the average and SD of three experiments. Significant differences are indicated in the figure as *, **, and ***, which correspond to *p* < 0.05, *p* < 0.01, and *p* < 0.001, respectively.

**Figure 4 cancers-14-03642-f004:**
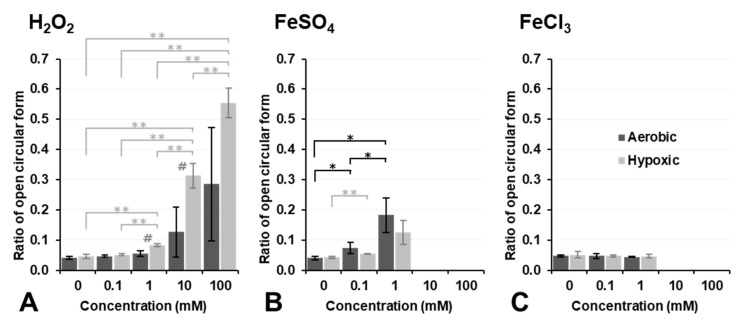
Estimation of plasmid DNA breakage induced by the administration of H_2_O_2_, FeSO_4_, or FeCl_3_ alone. Ratio of DNA breakage after reacting with (**A**) H_2_O_2_, (**B**) FeSO_4_, or (**C**) FeCl_3_. Dark and light columns indicate results of aerobic and hypoxic experiments, respectively. Columns and error bars indicate the average and SD of three experiments. Significant differences between different concentrations are indicated in the figure as * and **, which correspond to *p* < 0.05 and *p* < 0.01, respectively. # indicates a significant difference of *p* < 0.05 between aerobic and hypoxic.

**Figure 5 cancers-14-03642-f005:**
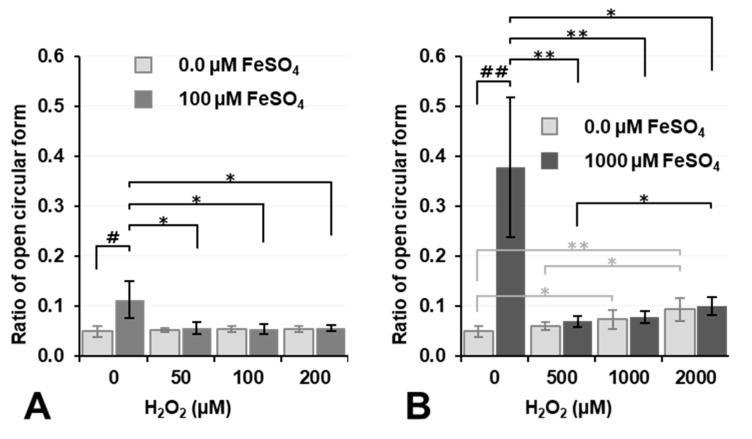
Estimation of plasmid DNA breakage induced by the Fenton reaction. (**A**) Ratio of DNA breakage after reacting 100 µM FeSO_4_ with several different concentrations of H_2_O_2_. (**B**) Ratio of DNA breakage after reacting 1000 µM FeSO_4_ with several different concentrations of H_2_O_2_. Columns and error bars indicate the average and SD of three experiments. Significant differences between different H_2_O_2_ concentrations are indicated in the figure as * and **, which correspond to *p* < 0.05 and *p* < 0.01, respectively. Significant differences between different FeSO_4_ concentrations are indicated in the figure as # and ##, which correspond to *p* < 0.05 and *p* < 0.01, respectively.

**Figure 6 cancers-14-03642-f006:**
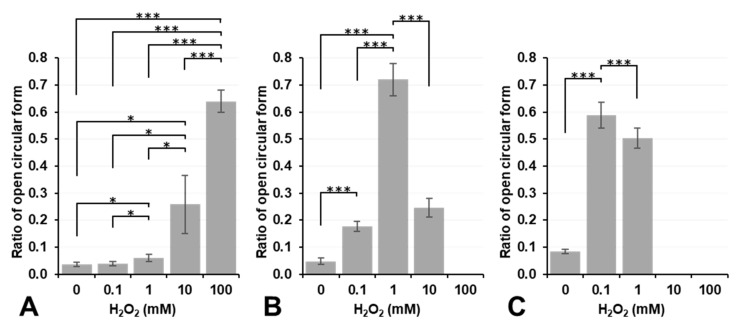
Estimation of plasmid DNA breakage induced by the ^•^OH generated by the decomposition of H_2_O_2_ under UV irradiation. Ratio of DNA breakage after UV irradiation at (**A**) 0.0, (**B**) 0.1, and (**C**) 0.5 J/cm^2^ with several different concentrations of H_2_O_2_. Columns and error bars indicate the average and SD of four experiments. Significant differences between different H_2_O_2_ concentrations are indicated in the figure as * and ***, which correspond to *p* < 0.05 and *p* < 0.001, respectively.

**Figure 7 cancers-14-03642-f007:**
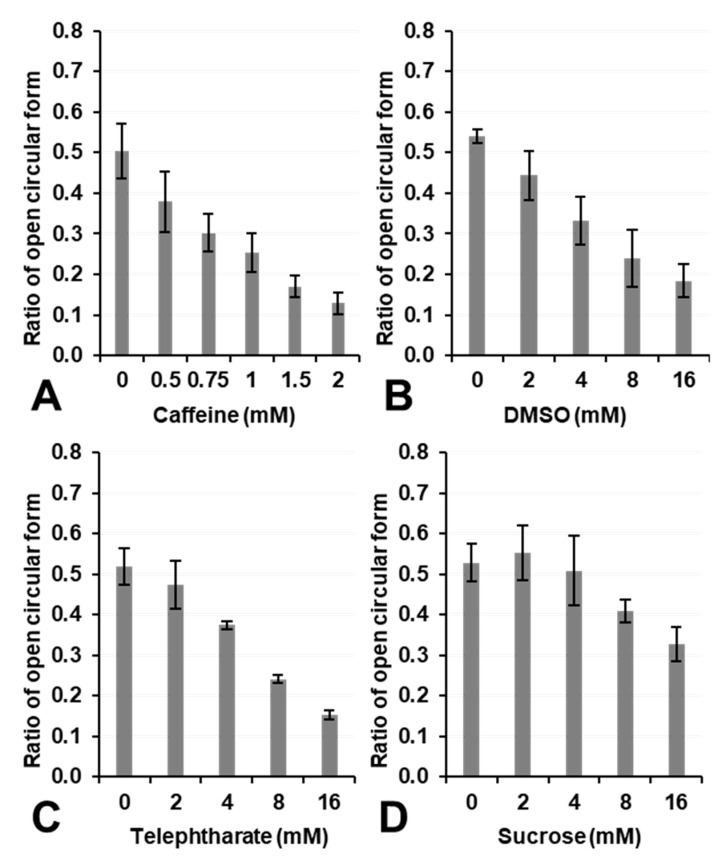
Effects of ^•^OH scavengers on plasmid DNA breakage induced by UV irradiation in samples containing H_2_O_2_. The ratio of DNA breakage was suppressed in a concentration-dependent manner by (**A**) caffeine, (**B**) DMSO, (**C**) terephthalate, and (**D**) sucrose. Columns and error bars indicate the average and SD of 3 experiments.

**Figure 8 cancers-14-03642-f008:**
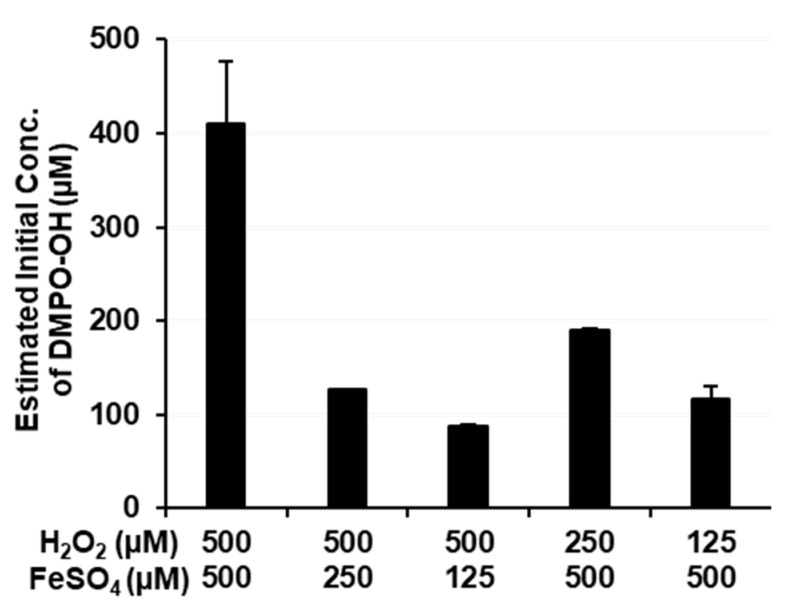
Estimation of the initial concentration of ^•^OH generated by the Fenton reaction. Columns and error bars indicate the average and SD of 3 experiments.

## Data Availability

The data presented in this study are available on request from the corresponding author.

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
