# Peer review of "Importance of Locations of Iron Ions to Elicit Cytotoxicity Induced by a Fenton-Type Reaction"

_cancers, 2022, doi:10.3390/cancers14153642_

Round 1

Reviewer 1 Report

Authors have addressed all of my concerns. May be accepted for the publication.

Reviewer 2 Report

The addition of new experiments increased the paper's quality, which is now sufficient for publication

This manuscript is a resubmission of an earlier submission. The following is a list of the peer review reports and author responses from that submission.

Round 1

Reviewer 1 Report

In their study, Igarashi and colleagues show that the locations of iron ions can affect the effects of a Fenton-type reaction on cellular cytotoxicity. They also found that the effects of the reaction are not caused by DNA damage. This is an interesting but incomplete study. The authors need to identify the factors that cause the cell death and toxic effects of the reaction. It is possible that the presence of Fenton's reagent activates certain signaling pathways. Hydroxyl radicals can cause oxidative stress to the cells and cause protein misfolding and aggregation. This may also increase cytotoxicity. Due to the incompleteness of the manuscript I cannot recommend it to be published in this journal.  

Reviewer 2 Report

I find the study to be  designed and performed cleverly, achieving an interesting general result with simple means. My only suggestion is to extend the quantitative analysis of OH* generation/recombination by trying to correlate intermolecular distances from the concentrations of reagents for all experiments with the extent of deleterious effect.

The authors did it in a couple of instances and I recommend that they do it more systematically.